# A Study of a Protein-Folding Machine: Transient Rotation of the Polypeptide Backbone Facilitates Rapid Folding of Protein Domains in All-Atom Molecular Dynamics Simulations

**DOI:** 10.3390/ijms241210049

**Published:** 2023-06-13

**Authors:** Harutyun Sahakyan, Karen Nazaryan, Arcady Mushegian, Irina Sorokina

**Affiliations:** 1Institute of Molecular Biology, Academy of Sciences of Republic of Armenia, Yerevan 0014, Armenia; harutyun_sahakyan@nih.gov (H.S.); karen.nazaryan@gmail.com (K.N.); 2Division of Molecular and Cellular Biosciences, National Science Foundation, Alexandria, VA 22314, USA; mushegian2@gmail.com; 3Strenic LLC, McLean, VA 22102, USA

**Keywords:** protein folding, ribosome function, molecular dynamics, energy-dependent protein folding, co-translational protein folding, nascent peptide rotation, peptide backbone manipulation, protein-folding machine, steered molecular dynamics

## Abstract

Molecular dynamics simulations of protein folding typically consider the polypeptide chain at equilibrium and in isolation from the cellular components. We argue that in order to understand protein folding as it occurs in vivo, it should be modeled as an active, energy-dependent process, in which the cellular protein-folding machine directly manipulates the polypeptide. We conducted all-atom molecular dynamics simulations of four protein domains, whose folding from the extended state was augmented by the application of rotational force to the C-terminal amino acid, while the movement of the N-terminal amino acid was restrained. We have shown earlier that such a simple manipulation of peptide backbone facilitated the formation of native structures in diverse α-helical peptides. In this study, the simulation protocol was modified, to apply the backbone rotation and movement restriction only for a short time at the start of simulation. This transient application of a mechanical force to the peptide is sufficient to accelerate, by at least an order of magnitude, the folding of four protein domains from different structural classes to their native or native-like conformations. Our in silico experiments show that a compact stable fold may be attained more readily when the motions of the polypeptide are biased by external forces and constraints.

## 1. Introduction

In the living cell, most proteins rapidly attain their native conformations while they are being synthesized, or shortly thereafter [1,2,3,4,5]. The folding process in vivo involves interactions of the polypeptide chain with the ribosome, with molecular chaperones, and with other maturation factors. In contrast, the folding process in vitro, in isolation from those cellular factors, can be accomplished only for small protein domains, and despite many decades of laboratory studies, theory development, and computer simulations, we still lack reliable methods of recapitulating the complete folding trajectories of most proteins in silico. The impressive success of machine-learning methods in predicting the static native conformations of proteins [6,7,8] does not extend to the prediction of protein-folding trajectories, and thus the problem of the protein-folding mechanism remains unsolved [9,10,11].

The lack of progress in learning the folding pathways for most polypeptides is sometimes attributed to the vast complexity of the problem. The number of possible conformations of a polypeptide chain, the number of interactions of the amino acid residues within the polypeptide and with the surrounding solvent, and the possibilities of interaction with other cellular components are all astronomically high, and calculation of physical parameters in such a system is still prohibitive, even with the modern computing power. We believe, however, that there are also fundamental deficiencies in our understanding of protein folding as it occurs in living cells, which limits our ability to develop realistic physical models of this process. Indeed, the predominant thermodynamic hypothesis of protein folding [12,13] has led to physical models that describe folding as a thermodynamically favorable, unassisted process, imagined as a protein molecule walk on a rugged funnel-shaped energy landscape between the conformations with decreasing Gibbs free energy, towards the global thermodynamic minimum at the bottom of the funnel [14,15,16,17,18,19,20]. We have argued that such models disagree with the extensive empirical data on protein solubility and stability, and with the modern understanding of the selective forces that have been acting on genetically encoded biopolymers during most of the evolution of life on Earth [21,22,23]. Alternative views on the mechanisms of protein folding in vivo must be explored.

We have proposed that more realistic physical models must consider the possibility that protein folding in vivo is an active, energy-dependent process [21,23]. A living cell is an open system with a constant flow of energy and with changing chemical composition. During protein synthesis on the ribosome, the nascent peptide emerges into a confined space of the ribosomal tunnel and then into a crowded molecular environment just outside the ribosomal exit. In the course of peptide synthesis and co-translational folding, a large amount of GTP is hydrolyzed by the translation factors, releasing energy that is not required for the formation of peptide bonds and is believed mostly to dissipate as heat [24]. We have argued that protein folding in vivo must account for the interactions of a folding polypeptide chain with its complex dynamic cellular environment, which is supplied with energy [21,22,23,25,26]. In this alternative model, proteins do not have to be able to fold spontaneously, instead they fold into their native conformations while being affected by the external forces applied by a protein-folding machine.

We hypothesized that the key role of the protein-folding machine is to manipulate the peptide backbone directly. This may be most readily achieved by the concerted actions of the ribosome and chaperone complexes on the polypeptide chain in the course of protein biosynthesis. During translation, the 3′ terminus of the tRNA in the A-site of the ribosomal peptidyl transferase center swings by nearly 180 degrees in every elongation cycle [27,28,29], and we have suggested that this motion may lead to the twisting of the C-terminus of the nascent polypeptide chain. Simultaneously, the movements of the N-terminal regions of the nascent peptides may be restricted by occlusions in the ribosome exit tunnel and then by steric capture mediated by the ribosome-associated chaperones [25]. As a result, the folding polypeptides may be placed into strained conformations with elevated free energy and reduced entropy; such states may then rearrange into the metastable conformations occupying local minima of free energy [23].

To explore the feasibility of a protein-folding machine facilitating the acquisition of native structure by mechanical manipulation of the peptide backbone, we have performed steered molecular dynamics simulations, augmented by application of torsion to the peptide backbones. In our experiments, directional rotation of the C-terminal amino acids with simultaneous limitation of the movements of the N-termini indeed facilitated the formation of native structures in five helical peptides [26]. Here we simulate the folding of the entire protein domains, and report that a transient twist of the protein backbone similarly enhances the formation of native structures in domains with different types of structure.

## 2. Results

### 2.1. Steered Molecular Dynamics with Transient Backbone Rotation

In an earlier work on helical peptide folding, we used all-atom molecular dynamics simulations in the explicit water environment, in which the simulation was augmented with a rotatory force applied to the C-terminal amino acid, while the motions of the N-terminal amino acid were restrained [26]. The five peptides started with the extended state and attained their native α-helical structure at least eight to ten times faster than when the folding was free, i.e., unassisted by rotation. Counterintuitively, the facilitation of folding was observed only when rotation was applied against the direction of the α-helix, suggesting that the peptide backbone does not passively follow the force vector but is undergoing a more complex, ordered rearrangement [26]. Thus, in the presence of the appropriately calibrated directional rotation force, the peptides do not behave as freely jointed chains; application of the force appears to be effective in reordering the peptides into folded conformations.

Here we asked whether external torque may facilitate folding of the entire protein domains that consist of several elements of secondary structure. We changed our protocol to apply the backbone steering at the start of simulation, and then continued the run without restraints. The force was applied during the first 250 ns of simulation, with the reference rate 0.36 degree/ps and a force constant 10,000 kJ·mol^−1^·nm^−2^, allowing us to perform smooth rotation of the C-terminal amino acid. As in our earlier experiments, we applied the rotation against the direction of the α-helix. After the transient 250 ns rotation, all forces were removed and the constraints were released. Control simulations were performed similarly to the rotation-augmented runs, but without restraints or manipulations of the polypeptide at any point. For each protein domain, simulations with transient rotation were run three times, and unconstrained simulations were also run in triplicate. To monitor the protein-folding process, we collected snapshots every 100 ps. At the whole-domain level, we monitored the root mean square deviation (RMSD) distance from the known native three-dimensional structure of the folded domain and the fraction of all native contacts formed by the amino acids within the domain. At the amino acid level, we tested whether the residue was incorporated into a correct element of the secondary structure and whether it formed a native contact.

### 2.2. Trp-Cage and BBA: Enforced Rotation of the Backbone Facilitates Rapid Acquisition of the Native Structure in Two Designed Single-Domain Proteins

For our simulations, we selected protein domains whose folding in vitro and in silico has been extensively studied. Two of those domains are proteins designed for efficient folding and stability in vitro [30,31]. They are the Tpr-cage domain variant (PDB ID 2JOF, 20 amino acids), which consists of a helix-coil-turn motif and a proline tail that packs against the helix, and the BBA domain variant FDS-EY (PDB ID 1FME, 28 amino acids), a derivative of Zn finger that is stable in the metal-free form and consists of two β-strands followed by an α-helix. Both peptides were placed in their simulation boxes in the stretched conformations, and both rapidly folded to their native structures when transient directional rotation was applied. The details of the folding process are presented in Figure 1 and Figure 2.

The Trp-cage domain folded in 0.5–1.5 µs, as can be seen from the time courses of the RMSD distances between the folding intermediate and the native structure (Figure 1A, left panel) and of the accumulation of the native contacts (Figure 1B, left panel). The folding times observed in our steered simulations are shorter by nearly an order of magnitude than the times of unassisted MD simulations (4–14 µs) reported in the literature for various versions of this domain including 2JOF [32,33,34,35]. An experimental study of the in vitro folding of the Trp-cage variant with the original sequence composition of the helix (PDB ID 1L2Y) also suggested 4 µs folding time [36]. In our control simulations, which were run for 5 µs, the Trp-cage domain did not fold (Figure 1A,B, right panels), though a fleeting native-like structure could be observed by sorting the frames by the RMSD distance from the target (see Appendix A for additional visualization).

Generally similar folding behavior was observed with a structurally different BBA domain; it acquired native structure within 1–2.5 µs from the start of the simulation (Figure 2A,B, left panels). The folding time for this domain with the transient rotation was also an order of magnitude shorter than 18 µs reported for the unassisted simulations [34,37]. In our control runs, we did not observe any stable compact structure for the 5 µs duration of the simulations (Figure 2A,B, right panels, and Appendix A).

A more detailed view of the folding process is provided by the analysis of secondary structures and native contacts at the residue level. The Trp-cage domain and BBA domain each contain a helical segment in their native structure—respectively, in the N-terminal and C-terminal region of the polypeptide chain. As can be seen in Figure 1C and Figure 2C, these helices rapidly formed at the initial stage of the simulation, when the backbone twist was applied. After the switch to unassisted simulation, the helices remained stable. The other parts of these domains, i.e., the C-terminal portion of the Trp-cage and the N-terminal portion of the BBA domain, did not form helices during the backbone rotation stage, and rapidly acquired their native conformations after the release from rotation. In the course of unassisted control simulations, the two polypeptides formed neither helices nor other elements of the secondary or tertiary structure. Substructures with helix-like or strand-like properties could be seen, but these states were occupied only transiently and many were located in the regions that have different type of structure in the native form of the protein (Figure 1C and Figure 2C).

For each target, the summaries of independent steered simulations can be compared to each other, providing visual information about the folding pathways (Figure 1D and Figure 2D). For the Trp-cage domain, the late stages of simulation are qualitatively very similar in all three runs, with the same residues forming the native contacts in the same order; the main difference is in the length of time between the end of the steered rotation phase and the start of the main folding event, and in the rate of folding (Figure 1D). The same is true of the BBA domain; specific residues formed their native contacts in nearly the same order each time, and the difference was mostly in the time shift of the entire sequence of events (Figure 2D). Thus, these two protein molecules did not explore alternative folding routes in our experiments—even if multiple folding pathways and intermediates exist for those targets, the steered simulation robustly directed the folding process into a specific pathway for each domain.

### 2.3. HP35 and NTL9-39: Rapid Tending to Stable Conformations with Some Deviations from the Native States

We studied folding of two naturally occurring protein domains, the HP35 fragment of chicken villin headpiece domain (PDB ID 1YU5, 34 amino acids) and a fragment of the N-terminal domain of ribosomal protein L9 from *Geobacillus stearothermophilus* (NTL9-39 K12M mutant; PDB ID 2HBA, 39 amino acids). The former domain is made of two crossed α-helices connected by a linker that includes two loops and the third helix. The latter domain consists of a β-sheet formed by three antiparallel β-strands; an α-helix is located between strands 2 and 3 along the peptide chain and is packed against one edge of the sheet. The folding of both domains has been studied in some detail [38,39,40,41,42,43,44]. We simulated folding of these domains in the same way as for the other two domains, except that for technical reasons, with NTL9-39 we only could use a box that was 10.5 nm long in the X axis direction (Appendix A), and therefore the starting peptide was not fully stretched (its calculated fully-stretched length is 12.8 nm).

In the rotation-assisted simulations, folding into compact forms was observed for both HP35 and NTL9-39 (Figure 3 and Figure 4). Examining the case of HP35 first, we see that in the unassisted runs this domain attained conformations with various extent of helical structure, but those structures mostly did not persist (Run 2, Figure 3C, right panel; Only HP35 transiently approximated the native fold without assistance (but see Appendix A for a representative snapshot of a fleeting native-like structure). When rotation was applied at the beginning of the simulation, in one run (Run 1, Figure 3C, left panel), all three helices were formed to give a nearly-native conformation, at least temporarily. Previously, we found that a peptide corresponding to the C-terminal helix of HP35 adopted the correct helical structure eight to ten times faster when rotational restriction was applied to its backbone than in the unassisted simulations [26]. When applied to the HP domain as a whole, a transient rotation of the peptide backbone improved its folding to a nearly-native conformation, though it did not strongly improve the persistence of such conformation (Figure 3).

Analysis of the free and assisted folding of the N-terminal domain of the ribosomal protein L9 revealed much more complex behavior. In the rotation-assisted simulations, NTL9-39 folded quickly, within 1–1.5 µs (orange and magenta curves in Figure 4, panels A and B). This is approximately 20-fold faster than the 29 µs folding rate reported for the same domain [34]. The main folding intermediate, in which amino acids 22–35 formed a helix, was observed in all three runs while the transient rotation was still applied. After the restraints and external forces were removed, in two runs the C-terminal portion of the helix (amino acids 30–35) relaxed into a loop, while the rest of the helix was retained, the β-strands were formed elsewhere in the molecule, and NTL9-39 attained compact conformation. Without assistance, NTL9-39 did not fold into a compact structure for the duration of the simulations (Figure 4A,B, right panels, and Appendix A).

The structure that was formed in the rotation-assisted runs and remained stable for most of the simulation displayed interesting local deviations from the native conformation of NTL9-39. Focusing on the run 2, we note that the majority of the fold, i.e., the helix and strands 1 and 2 that directly interact with it, formed correctly. The third β-strand completing the β-sheet was also observed. The examination of snapshots in the course of the simulation, however, revealed a dynamic transition between two distinct conformations. One conformation was nearly identical to the native structure (Figure 4E, left), but existed only for 0.5 µs. For a longer period, 1.2 µs in total, the direction of the third strand within the fold was reversed from antiparallel to parallel, which required the extreme C-terminus of the α-helix to frazzle and the connecting loop to stretch over the edge of the folded domain (Figure 4E, right).

The alternative structure had relatively short RMSD distance from the database NTL9-39 structure—only slightly higher than the nearly-native form of NTL9-39—as many backbone atoms in the flipped strand could be still be superimposed onto the native form (Figure 4A). However, another domain-level measure, the fraction of native contacts Q, was relatively low for the alternative form, because the contacts formed by the third strand and by the reorganized loop of NTL9-39 were different from the native ones (Figure 4B). Similarly, at the amino acid residue level, most residues were incorporated into the correct type of the secondary structure (Figure 4C); at the same time, many amino acids in the third strand, the preceding loop, and their interacting residues formed the new, non-native contacts (Figure 4D).

## 3. Discussion

In an earlier work, we began testing the idea that addition of external forces to a molecular dynamics simulation can improve modeling of protein-folding pathways in silico [26]. Steered molecular dynamics approaches, in which physical forces and the motion restrictions are imposed on the system, have been explored before, mostly for the analysis of protein unfolding and for the studies of relatively small structural changes in the folded proteins, such as allosteric rearrangements induced by ligand binding [45,46,47,48]. We are applying steered MD for a different purpose, i.e., to study the rate and the pathways of protein folding de novo.

The domains that were folded in this work consist of multiple secondary structure elements; all domains include α-helices, and two of the domains also contain β-strands. For each of the targets, steered simulations resulted in the domain folding that occurred much faster than in unassisted simulations, confirming that rotation of the peptide backbone can have significant consequences for the folding dynamics of the entire protein domains.

Two domains, Trp cage and BBA, folded to their native conformations when a transient rotation was applied to the polypeptide chain, but did not fold unassisted, most likely because of insufficient simulation time on the commodity hardware available to us. Both domains have been designed for their high stability in solution, but their folding speed appears to be significantly enhanced by transient backbone rotation described here. In both cases, this acceleration of folding is achieved on a single dominant pathway, with specific residues forming native contacts in the same relative order, without visibly populating alternative conformations. This is in contrast with the data on the unassisted folding of these domains from the D.E. Shaw group, where the order of formation of the native contacts differed significantly between simulation runs (see Figure 3 in [34]).

With the other two domains, folding to the compact conformation was also accelerated by the transient rotation of the polypeptide, but instead of settling into the native topology, the polypeptides assumed alternative conformations. The compact conformation sharing two of the three helices with the native fold of HP35 was attained inside the 5 μs of simulation only when transient rotation was applied to the peptide backbone. The folding of the villin headpiece domain HP35 has been postulated to occur at its “physical speed limit” in vitro and in silico [38,39], but our results suggest that when a rotatory external force is applied to the backbone, such a barrier may be crossed.

Folding of the NTL9 domain also benefited from assistance, and, interestingly, the folded molecule visited two alternative compact forms. One of those was the native three-stranded antiparallel β-sheet, but it was populated only about 12% of the simulation time. The other form, occupied for almost 25% of the simulation, contained a β-sheet with alternative topology. Notably, despite considerable work on the folding of the 39-residue NTL9 domain in vitro, the three-dimensional structures of NTL9 are known only for a 52-residue version. The 52-residue form includes an additional long C-terminal α-helix, which is likely to preclude the last β-strand from flipping into an alternative position during isolation and crystallization. We speculate that the smaller NTL9-39 domain, which lacks this helix, may fold either into the same form, or the alternative form observed by us, or perhaps is represented by a mixture of both forms in vitro.

The non-native conformations are sometimes observed in the MD simulations of protein domain folding, including the four targets discussed in this work [41,43,49,50]. We believe that such alternative conformations are in fact quite common, but when the “correct” conformation is known in advance, the “wrong” conformations may be seen as technical artifacts, or perhaps as occasional kinetic traps in the rugged folding landscapes [50]. On the contrary, we believe that the metastable alternative conformations of protein domains deserve special attention, because they may provide clues to the mechanisms of protein folding and misfolding in vitro and in vivo. In addition, the discrepancy between the readouts that are commonly used to monitor the fit to the known structure, such as the one observed in the case of NTL9-39, is worth keeping in mind when analyzing the results of molecular simulations.

Our results may have implications for the understanding of the protein-folding mechanisms in vivo as well. We have postulated the importance of the cellular protein-folding machine, i.e., the active folding of the nascent proteins by the cellular systems, such as ribosomes in concert with molecular chaperones, which may directly alter the conformations of nascent proteins by applying mechanical force to the peptide backbone [21,22,23]. The feasibility of such a process depends on whether the manipulation at one site of a peptide could propagate through the rest of the peptide chain and affect the movements of the distal sites. In an earlier study, we used a steered MD simulation to show the conditions under which the peptides indeed did not behave as freely jointed chains: when a rotation force was applied to a single amino acid residue, and the motion of just one distal residue was restricted, the folding trajectory of the entire peptide was affected dramatically, leading to the rapid attainment of the native helical conformation [26]. Here we show that a transient application of the rotation force at the beginning of the simulation is sufficient to accelerate folding of four diverse protein domains. Apparently, after the initial short application of the force, the protein chains remain in a state that is not freely jointed, and a transient manipulation leads polypeptides into strained conformations with elevated free energy, from which they transition to stable native-like conformations. In the case of NTL9-39, moreover, our MD revealed a non-native conformation that is more stable, at least kinetically, than the purported native structure of this domain; it would be of great interest to know whether these structures are more stable or less stable than the unfolded form.

To conclude, we are convinced that protein folding should be simulated in silico in such a way that recapitulates the folding process as it actually occurs in vivo, i.e., as a non-equilibrium, active, energy-dependent process [23]. We believe that the evidence of the effects of rotational restriction of the backbone that we are exploring in our simulations may help to introduce some realism into modeling of protein folding.

## 4. Materials and Methods

Molecular dynamics simulations were carried out using the GROMACS-2020.4 package with CHARMM36m force field (July 2020 release) [51,52]. The initial unfolded structures of proteins were produced with the ICM-Pro program (MOlSoft LLC, San Diego, CA, USA) [53]. The initial unfolded structures were generated by setting the phi and psi dihedral angles to 180 degrees, and a local energy minimization was performed within ICM-Pro to relax the dihedrals and side chains. After this treatment, peptides remained in an extended linear form and did not possess any local secondary structure. The peptide backbones were aligned along the X-axis and solvated in triclinic boxes with TIP3P water model and potassium and chloride ions. The system was then minimized with the steepest descent algorithm and equilibrated with NVT/NPT ensembles for 1 ns at each equilibration cycle. The MD simulations were performed at a constant temperature of 300 K with the v-rescale thermostat and a time constant of 0.1 ps. The pressure was coupled at 1 bar with Parrinello–Rahman barostat using a time constant of 2 ps [54,55,56]. Electrostatic forces were calculated using the particle-mesh Ewald method with a Fourier grid spacing of 0.16 nm. Short-range Van der Waals interactions were force-switched off from 0.8 to 1.2 nm [57]. The bond lengths were constrained using the LINCS algorithm [58], and a 2 fs integration time step was used. Snapshots were collected every 100 ps.

The rotation of the C-terminal amino acid was performed using the enforced rotation method implemented in Gromacs [59], as repurposed in our previous study [26]. The rotation was applied during the first 250 ns to the C-terminal amino acid using the flexible axis approach (Vflex2) with a reference rotation rate of 0.36 degree/ps (1 rotation per 1 ns) and force constant 10,000 kJ·mol^−1^·nm^−2^, and continued without restraints and without applied rotation thereafter. Control simulations were performed without any initial restraint or rotation. The properties of the simulation boxes for every run are given in the Appendix A Appendix A.

The RMS distances from the database-deposited native structure were calculated in Gromacs. Acquisition of secondary structure and the fraction of native contacts were assessed using VMD software (version 1.9.3) [60]. VMD was also used for trajectory visualization. Native contacts for an amino acid were considered to be formed if that amino acid has the same contacts as in the reference X-ray structure within a cutoff of 8 Å. The fraction of native contacts (Q) was calculated as Q = N/N_all_, where N is the number of the native contacts at a given time and N_all_ is the number of contacts in the reference X-ray structure.

## 5. Patents

I.S. is an inventor on a patent application that has been filed in connection with the methods described in this article.

## Figures and Tables

**Figure 1 ijms-24-10049-f001:**
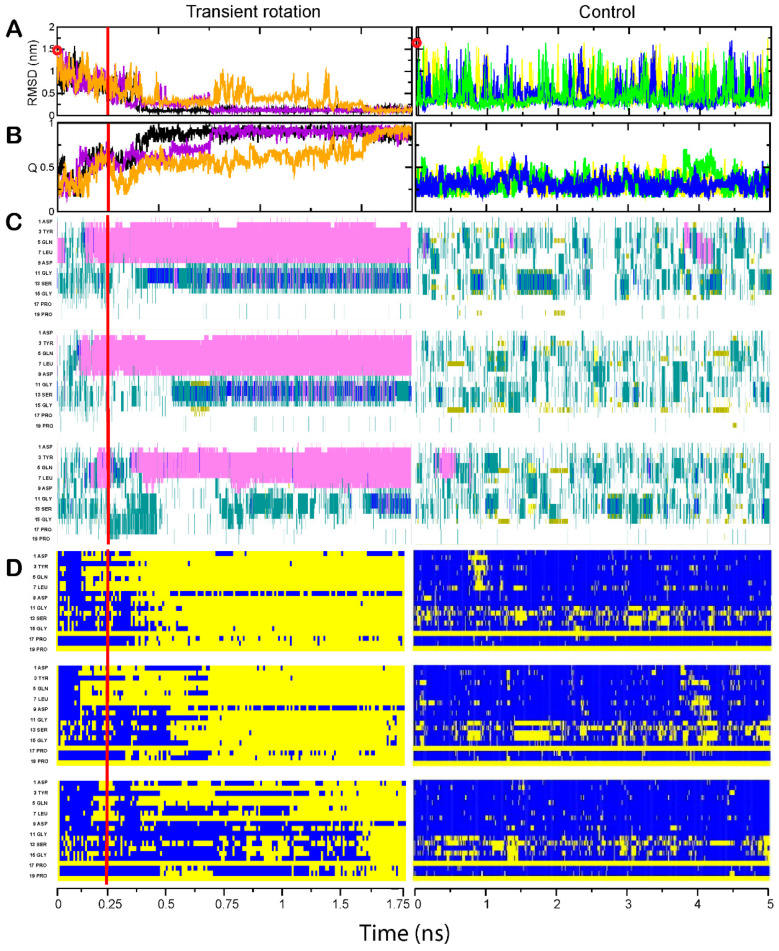
Folding of the Trp-cage domain. In each of the panels (**A**–**D**), three simulations performed with transient rotation of the C-terminal amino acid are shown in the left column, and the three control unassisted simulations are depicted in the right column. In panels (**A**,**B**), each run is marked in a different color. The time of simulation in microseconds is indicated on the abscissa underneath each column, and red vertical bars in the left column indicate the end of transient rotation and relief of spatial constraints. (**A**) RMSD distance of protein backbone from the folded state. The open red circles on the Y axis indicate the calculated RMSD values at the start of simulations. (**B**) The fraction of native contacts Q attained by the molecule. The colors of the trajectories match those in (**A**). (**C**) Inclusion of each residue into a secondary structure element. The identity and position of every other residue are shown on the Y axis. α-helices are shown in magenta, 3_10_ helices in blue, β-strands in yellow, turns in green, and coils in white. (**D**) Per-residue formation of the native contacts. Blue indicates lack of the native contact; yellow indicates that the native contact is formed.

**Figure 2 ijms-24-10049-f002:**
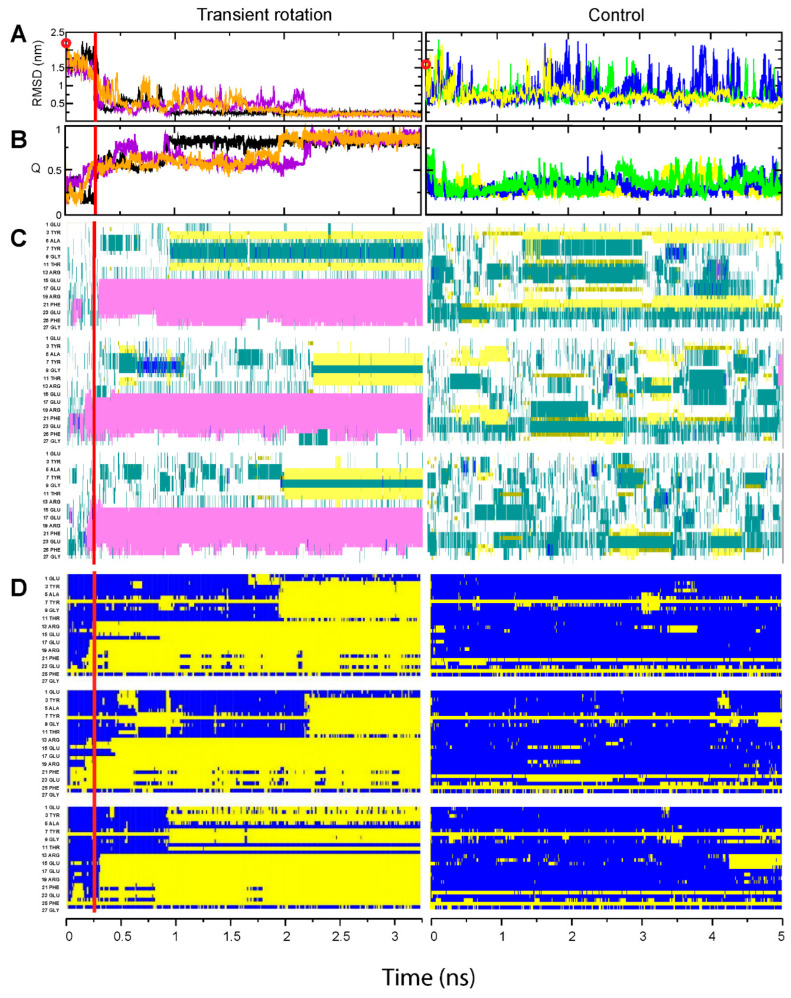
Folding of the BBA domain. Panel designations and color codes are the same as in Figure 1. In each of the panels (**A**–**D**), three simulations performed with transient rotation of the C-terminal amino acid are shown in the left column, and the three control unassisted simulations are depicted in the right column. In panels (**A**,**B**), each run is marked in a different color. The time of simulation in microseconds is indicated on the abscissa underneath each column, and red vertical bars in the left column indicate the end of transient rotation and relief of spatial constraints. (**A**) RMSD distance of protein backbone from the folded state. The open red circles on the Y axis indicate the calculated RMSD values at the start of simulations. (**B**) The fraction of native contacts Q attained by the molecule. The colors of the trajectories match those in (**A**). (**C**) Inclusion of each residue into a secondary structure element. The identity and position of every other residue are shown on the Y axis. α-helices are shown in magenta, 3_10_ helices in blue, β-strands in yellow, turns in green, and coils in white. (**D**) Per-residue formation of the native contacts. Blue indicates lack of the native contact; yellow indicates that the native contact is formed.

**Figure 3 ijms-24-10049-f003:**
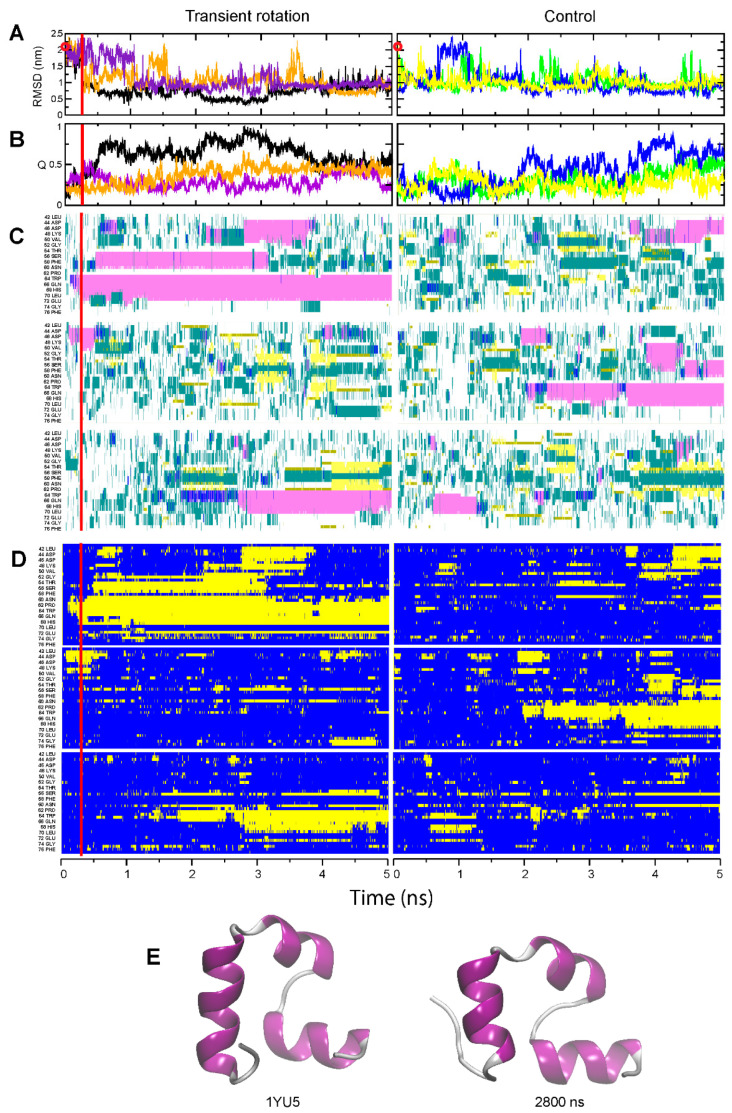
Folding of the HP35 domain. The panel designations and color codes for (**A**–**D**) are the same as in Figure 1 and Figure 2. (**E**) Left, native structure from the pdb entry 1YU5; right, transiently observed nearly-native structure (snapshot at 2.8 µs) from the assisted Run 2 (purple curves in panels (**A**,**B**)).

**Figure 4 ijms-24-10049-f004:**
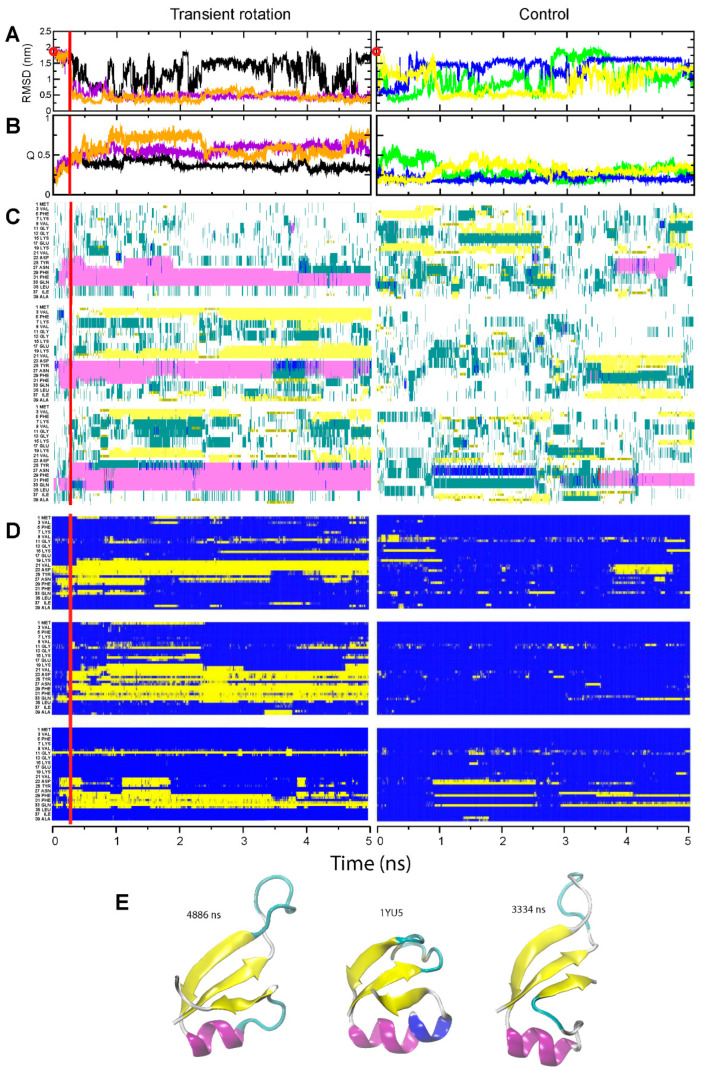
Folding of the NTL9-39 domain. The panel designations and color codes for (**A**–**D**) are the same as in Figure 1, Figure 2 and Figure 3. (**E**) The native structure of NTL9-39 and two alternative compact conformations observed in steered Run 2 (purple curves in panels **A**,**B**): center, native structure derived from the pdb entry 2HBA; left, less frequently observed nearly-native structure (snapshot at 4.9 µs); right, more frequently observed alternative fold (snapshot at 3.3 µs).

## Data Availability

All MD simulation trajectories are available at https://doi.org/10.5281/zenodo.7882283, accessed on 11 June 2023.

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
