# Peer review of "A Study of a Protein-Folding Machine: Transient Rotation of the Polypeptide Backbone Facilitates Rapid Folding of Protein Domains in All-Atom Molecular Dynamics Simulations"

_ijms, 2023, doi:10.3390/ijms241210049_

Round 1
Reviewer 1 Report
This is an exciting paper and it is true that we need to simulate protein folding in an environment as close to possible as we can get to what happens within the cell, and this approach allows us to step closer to performing such simulations. However, the question I kept asking while reading this paper was if the outcome of these simulations, other than the folded structures, did match up with any experimental data? However, if the premise that this approach provides folding pathways closer to in vivo that might not match in vitro studies, then this may be of limited value, but would be interesting to know if the pathways obtained using this method do agree with experimental folding data.
Regarding the question of whether different pathways are follwed in vivo vs in vitro, the authors comment on alternative conformations and pathways, but no comparisons to others data are shown. It is clear that the formation of secondary structure proceeds more rapidly compared to the control simulations, but I think an analysis of the key stages in the folding mechanisms, for example intermediate structures, in more detail compared against that of other folding studies (MD or expt) on these proteins would make this paper more impactful. If possible, including more detail on this would be valuable and aid reflection on how we continue to study protein folding. It may be that such an analysis is more suited for a follow-up paper.
How unfolded were the starting structures? How does ICM-Pro generate the unfolded structures? Is this from simple chain building with no secondary structure propensities? A bit more detail in the methods would be appreciated to avoid the reader having to dig through previous papers for this information.
Unfortunately, all the figures are difficult to read. There is a lot of data here. I recommend these be made larger by having them as a single panel width or on landscape pages so that the axes and detail are more readable. The structures in some figures have numbers next to them for the timeframe but no units, I would recommend these units are added to the images and not just listed in the figure legend.
I think it would be beneficial if the figures showing the folded structures also included structures of where the control simulations got to in the same length of time. From the analysis shown this appears to be quite pronounced and I think just showing the structures as well would be a really great visual of the strength of the approach being presented here.
Reviewer 2 Report
This manuscript is well written and pleasant to read. I like the idea that taking the external force into the consideration of the protein folding process. As we know, the real situation in cell is far more complicated. It’s a super-crowded small case with a lot of interactions/regulations going on. Although the authors’ model is simplified, it’s a non-trivial exploration. The authors demonstrated that a transient rotational force can dramatically accelerate the folding of proteins in various forms. But I still have some questions for the authors as below:
1. How does the simulation protocol adopted in this manuscript mimic real situation?
2. A related question as above, how 250 ns was chosen at the beginning of the simulations? Is this time length reasonable?
3. The rate was 0.36 degree/ps. Have the authors tried other values? Would they produce very different results?
4. Similarly, how the force 10000 kJ mol-1 nm-2 was decided? How the magnitude of force affect the folding process?
5. The authors chose to apply the force against the direction of alpha-helix. There are beta-strands in the structure. How will the relative direction of the force to the beta-strands impact the folding?
6. The model protein domains were published structures. And, each simulation was repeated three times. I’d suggest the authors to show the comparison of the folded structures from three repetitive simulations and the published structure.
Round 2
Reviewer 2 Report
The authors have addressed my questions.